# Evaluating Open-Source Med-VLMs for Histopathologic Interpretation with Clinical Context

**Amir Nazem**[1]                                            ANazem@mdanderson.org
**Xiao Li**[2]                                                xiao.li@vanderbilt.edu
**Feng Yin**[1]                                               FYin@mdanderson.org
**Shilin Zhao**[2]                                            shilin.zhao.1@vumc.org
**Yang Ding**[3]                                              Yang.Ding@bmc.edu
**Haichun Yang**[4]                                           haichun.yang@vumc.org
**Yuankai Huo**[2]                                            yuankai.huo@vanderbilt.edu
**Yaohong Wang**[1]                                           yaohongwang@mdanderson.org

[1] *The University of Texas MD Anderson Cancer Center, Houston, TX, USA*

[2] *Vanderbilt University, Nashville, TN, USA*

[3] *Baylor College of Medicine, Houston, TX, USA*

[4] *Vanderbilt University Medical Center, Nashville, TN, USA*

## Abstract

The integration of artificial intelligence into pathology has driven research interest in evaluating medical vision-language models (Med-VLMs) for histopathologic interpretation. In this study, we compared two open-source Med-VLMs, MedGemma-4B-IT and Qwen2.5-VL-7B, for their ability to generate diagnostic interpretations from histopathology images with and without accompanying clinical context. A total of 100 cases were curated from the Pathology Outlines question bank and reviewed by two pathologists to generate controlled inputs, including multiple-choice differentials and removal of image-descriptive cues. Each model was evaluated across four experimental conditions with varying input modalities. Model outputs (diagnosis and reasoning) were scored by a pathologist on a 0–4 scale, and performance differences were analyzed using chi-square testing. Both models demonstrated significantly improved performance with the addition of clinical context and/or structured differential diagnoses ($\chi^2 > 10$, p < 0.001). The greatest improvement was observed when multiple-choice differentials were provided (MedGemma: +45.9%; Qwen: +35.7%), while clinical context alone yielded more modest gains (MedGemma: +30.6%; Qwen: +24.5%). Qwen exhibited greater robustness to inconsistent clinical information, with smaller performance declines under conflicting inputs (11.2% vs. 23.5%). These findings highlight the importance of structured contextual inputs in enhancing diagnostic performance of Med-VLMs and support the potential of open-source models for privacy-preserving, locally deployable AI systems in pathology.

**Keywords:** Artificial intelligence (AI), foundation models, medical vision-language models

## 1. Introduction

Artificial intelligence has become increasingly integrated into medical imaging workflows. In histopathology, conventional deep learning approaches are largely image-centric, whereas real-world diagnostic practice requires integration of morphologic features with clinical context and reasoning. The emergence of medical vision-language models (Med-VLMs) has introduced new opportunities for multimodal interpretation, enabling joint analysis of

histopathologic images and associated clinical information (Ding et al., 2025; Li et al., 2026). Recently, pathology foundation models have demonstrated promising diagnostic capabilities through large-scale multimodal learning (Wang et al., 2024); however, the relative contribution of visual and language inputs to model performance remains incompletely understood.

In this study, we systematically evaluate how visual and language inputs influence diagnostic reasoning in two well-established open-source Med-VLMs, MedGemma-4B-IT and Qwen2.5-VL-7B, using controlled experimental conditions with varying input modalities.

## 2. Methods

A dataset of 100 histopathology cases was curated from the publicly available Pathology Outlines question bank (Pathology outlines, 2025) (Sections: Neuropathology, Genitourinary and Gastrointestinal, 33-34 cases each). Each case consists of a representative field-of-view image and associated clinical context. Two pathologists reviewed all cases to standardize inputs by generating multiple-choice differential diagnoses and removing explicit image-descriptive cues to prevent bias (see Figure S1).

Two open-source medical vision-language models, MedGemma-4B-IT (see Figure S2) and Qwen2.5-VL-7B (see Figure S3), were evaluated under four controlled experimental conditions designed to isolate the contributions of visual and language inputs: (A) image-only, (B) image with clinical context, (C) image with structured differential diagnosis (multiple-choice options), and (D) combined input including both clinical context and differential diagnosis (see Figure 1.A and Figure S1).

For each condition, models were prompted to generate a primary diagnosis along with free-text reasoning. All outputs were independently reviewed by a pathologist and assigned a composite accuracy score on a 0–4 scale, incorporating both diagnostic correctness and quality of reasoning (see Table 1) (Li, et al 2024). Statistical analysis was performed using chi-square testing to compare score distributions across experimental conditions and between models.

| Score | Scenario |
|-------|----------|
| 0 | No answer provided |
| 1 | Incorrect diagnosis and incorrect reasoning |
| 2 | Either incorrect diagnosis or incorrect reasoning |
| 3 | Correct diagnosis, incomplete reasoning |
| 4 | Correct diagnosis, appropriate reasoning |

Table 1: Four-tier scoring scale for histopathological diagnosis

## 3. Results

Both MedGemma and Qwen demonstrated significantly improved interpretation scores with the inclusion of clinical context and/or structured differential diagnoses ($\chi^2 > 10$, p <

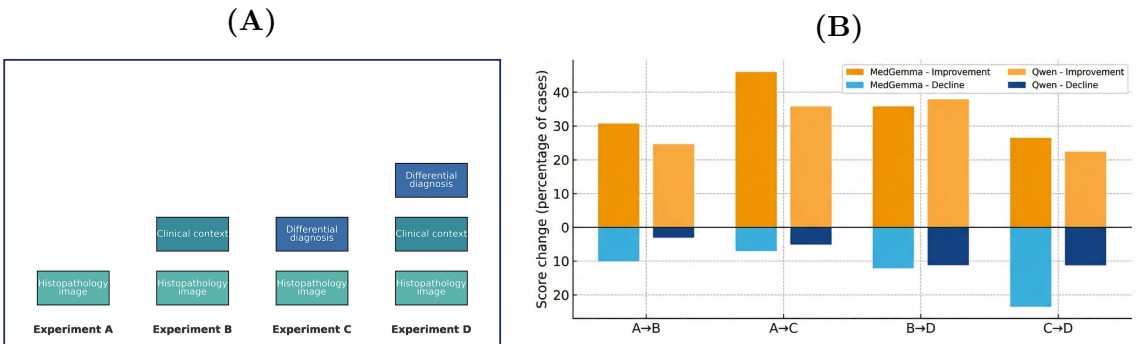

Figure 1: Study design (A). Diagnostic score changes, after including clinical context and/or differential diagnosis (B)

0.001). Across both models, the greatest performance gains were observed when multiple-choice differential diagnoses were provided (MedGemma: +45.9%; Qwen: +35.7%), whereas the addition of clinical context alone resulted in more modest improvements (MedGemma: +30.6%; Qwen: +24.5%) (see Figure 1.B). Notably, Qwen demonstrated greater robustness to inconsistent or distracting clinical information, with smaller performance declines between conditions C and D (MedGemma: 23.5% vs. Qwen: 11.2%).

## 4. Discussion

Our findings demonstrate that MED-VLM model performance is strongly influenced by textual inputs, particularly structured diagnostic prompts. Although vision-language models are designed to integrate visual and linguistic information, our results suggest that, at inference, language-based inputs play a dominant role in shaping diagnostic output. Of note, greater performance gains were observed with multiple-choice diagnostic prompts compared to clinical context alone. While the addition of clinical context improved diagnostic accuracy, its effect was comparatively modest, indicating that models may be more sensitive to constrained, task-specific prompts than to broader contextual narratives. These findings have important implications for the deployment of Med-VLMs in pathology. First, they highlight prompt design as a critical determinant of model performance, suggesting that structured inputs may enhance reliability in clinical decision support settings. At the same time, this reliance on language raises concerns regarding potential bias, as model outputs may be disproportionately influenced by how information is framed rather than purely by morphological characteristics identifiable on histology images. The observed differences in robustness between models, particularly under inconsistent or distracting inputs, further underscore the need to evaluate stability and generalizability in real-world scenarios.

Overall, this study highlights the diagnostic potential as well as current limitations of open-source Med-VLMs for histopathologic interpretation and their capacity to integrate clinical and morphologic information (Lu et al., 2024). Our findings suggest that future work should focus on improving image-driven reasoning while maintaining robustness to variations in clinical context.

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

## Appendix A. Supplemental Figures

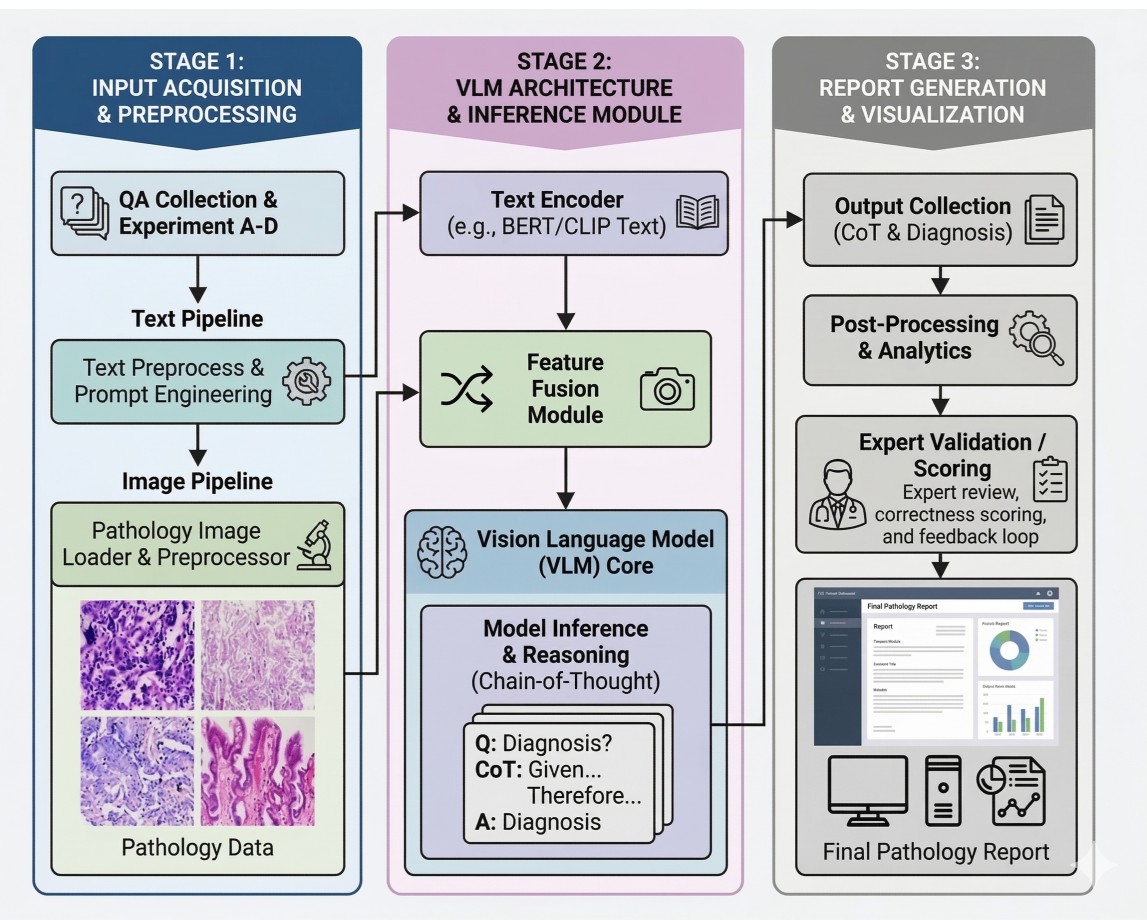

Figure S1: Methodology. We used a unified, scalable evaluation pipeline for multimodal medical reasoning, applying two cutting-edge vision-language models—MedGemma-4B-IT and Qwen2.5-VL-7B. Image inputs underwent standardized preprocessing, fusion, and normalization to ensure embedding-level consistency across models. Outputs were post-processed for comparative analysis across dimensions of diagnostic accuracy, reasoning fidelity, and vision-language alignment.

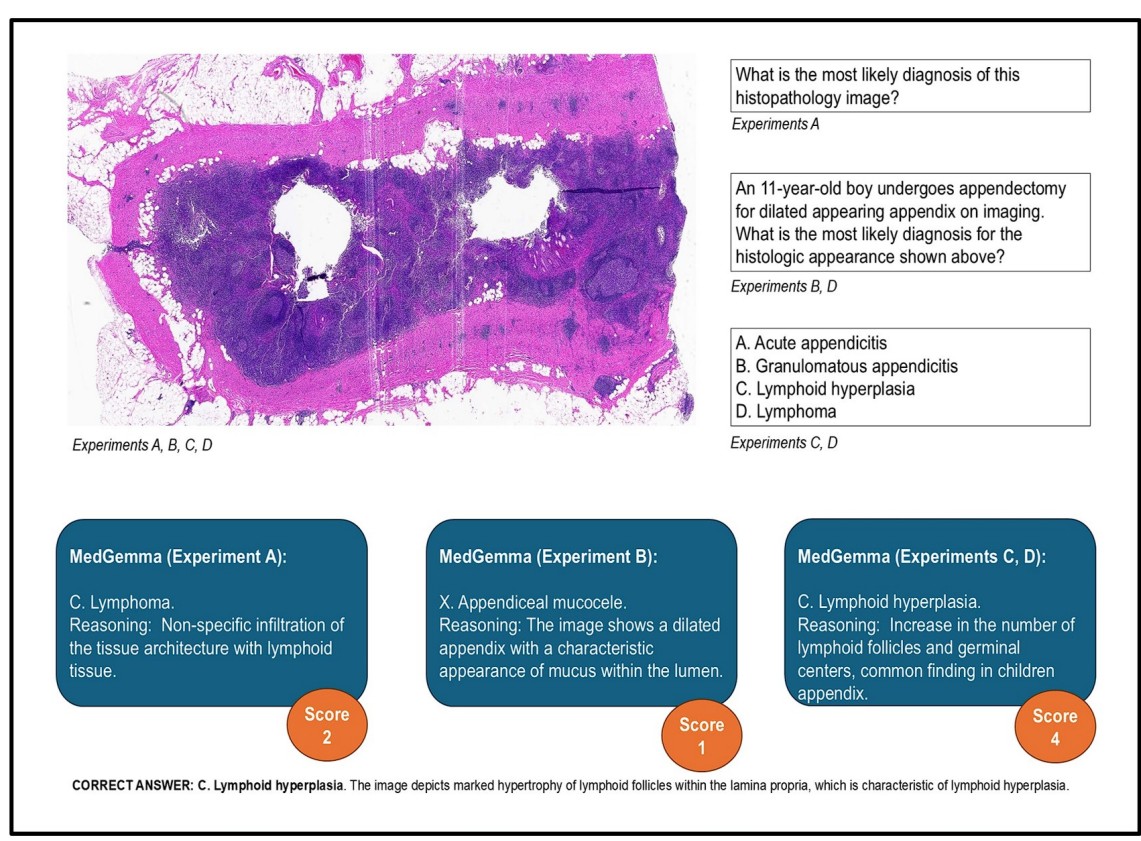

Figure S2: An example of experiments A-D with MedGemma model

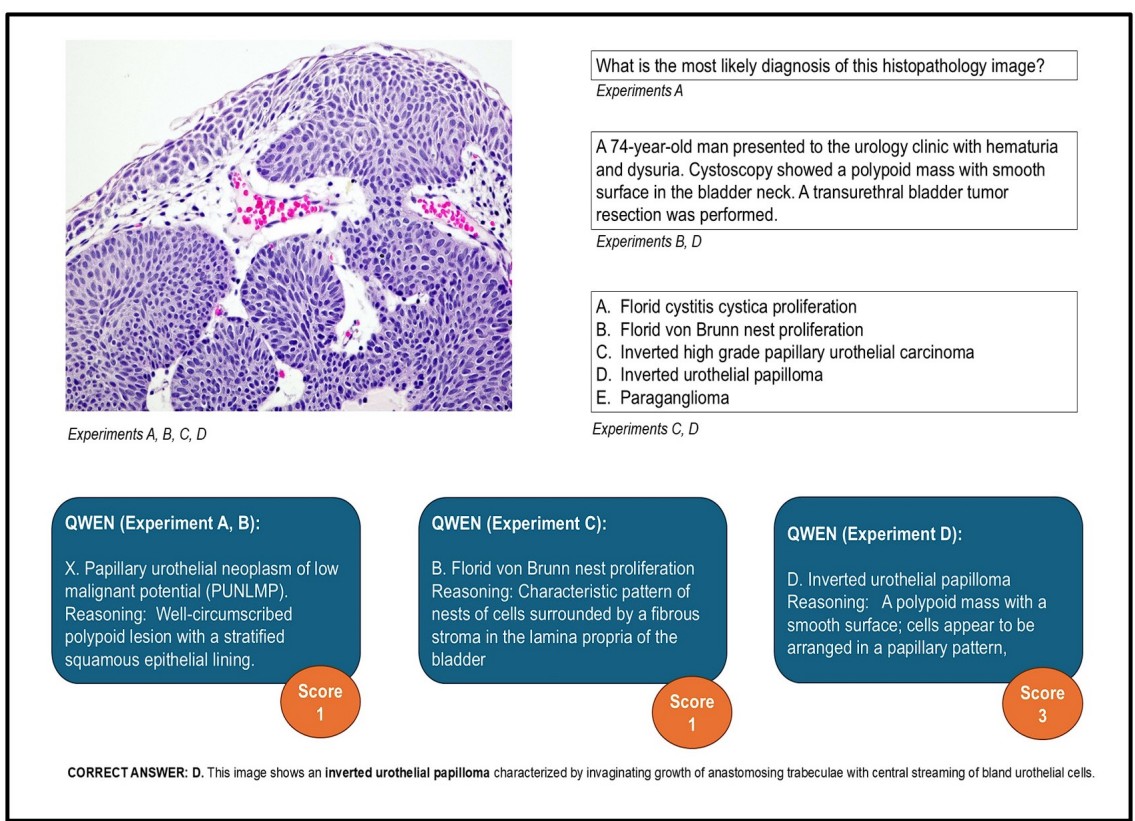

Figure S3: An example of experiments A-D with QWEN model

