# OpenReview forum: "Evaluating Open-Source Med-VLMs for Histopathologic Interpretation with Clinical Context"
_MIDL.io/2026/Short_Papers — MIDL 2026 - Short Papers Poster_

### Official Review · Reviewer_NsYx · 2026-05-03
**Weak Accept: A useful short evaluation showing Med-VLMs are highly sensitive to clinical context and structured prompts.**

**Rating:** 4
**Confidence:** 5

**Review:**

The study is clear, interpretable, and appropriate for a short paper. However, the contribution is mainly observational rather than methodological. The dataset is relatively small and curated from a pathology question bank; although the cases are clinically grounded, this format may not fully reflect the ambiguity and variability of real-world diagnostic practice. In addition, the multiple-choice setting may overestimate diagnostic reasoning ability compared with open-ended diagnosis. The paper would benefit from clearer score distributions, exact statistical reporting, and inter-rater agreement for pathologist scoring.

**Summary:**

The paper addresses a timely and important question: how open-source Med-VLMs perform on histopathology interpretation with and without clinical context. The controlled input conditions are useful, and the results suggest that structured text prompts, especially multiple-choice differentials, substantially improve model scores.

**Strengths:**

Its main strength is the controlled comparison of image-only, image + clinical context, image + structured differential diagnosis, and combined inputs, which provides a clear observation: current Med-VLM performance is strongly influenced by textual context, especially multiple-choice differential diagnoses. This is an important and practically relevant finding for pathology AI, because it highlights both the potential and the risks of using prompt/context-driven VLMs in diagnostic workflows.

**Weaknesses:**

Although the paper provides a scoring rubric and reports chi-square significance, the statistical reporting could be clearer. Full score distributions, exact test statistics, effect sizes, and inter-rater agreement for output scoring would strengthen the evaluation.

**Justification Of Rating:**

Overall, this is a relevant and useful short evaluation study.

---

### Decision · Program_Chairs · 2026-05-08

Accept (Poster)